# ‘Compressed Baby Head’: A New ‘Abusive Head Trauma’ Entity?

**DOI:** 10.3390/children10061003

**Published:** 2023-06-02

**Authors:** Enrica Macorano, Mattia Gentile, Giandomenico Stellacci, Mariano Manzionna, Federica Mele, Mariagrazia Calvano, Mirko Leonardelli, Stefano Duma, Giovanni De Gabriele, Alessandro Cristalli, Raffaella Minella, Aldo Di Fazio, Francesco Introna

**Affiliations:** 1Section of Legal Medicine, Interdisciplinary Department of Medicine, University of Bari ‘Aldo Moro’, 70124 Bari, Italy; f.mele17@studenti.uniba.it (F.M.); mariagrazia.calvano@uniba.it (M.C.); mirko.leonardelli@uniba.it (M.L.); stefano.duma@uniba.it (S.D.); g.degabriele@studenti.uniba.it (G.D.G.); a.cristalli@studenti.uniba.it (A.C.); francesco.introna@uniba.it (F.I.); 2Medical Genetics, Maternal and Child Department, Hospital of Venus, 70012 Bari, Italy; mattiagentile@libero.it; 3Unit of Paediatric Imaging, Giovanni XXIII Hospital, 70126 Bari, Italy; giandomenico.stellacci@policlinico.ba.it; 4Complex Operating Unit, Paediatric and Neonatology, San Paolo Hospital, ASL Bari, 70100 Bari, Italy; mariano.manzionna@asl.bari.it; 5School of Biological and Environmental Sciences, Liverpool John Moores University, Liverpool L3 5UX, UK; raffaella.minella@gmail.com; 6Regional Complex Intercompany Institute of Legal Medicine, 85100 Potenza, Italy; aldo.difazio@ospedalesancarlo.it

**Keywords:** child abuse and neglect, child maltreatment, physical abuse, abusive head trauma, shaken baby syndrome, shaken impact syndrome, compressed baby head

## Abstract

Background: *Child abuse* represents an important issue in the medico-legal and social context. In the last few decades, various aspects and mechanisms have been identified in *child abuse* case studies; however, constant research is needed in the field. With this paper, the authors will present a case of a new entity of *Abusive Head Trauma* that has come to the attention of medico-legal experts. Discussion: The trauma analysis performed on the cranio-encephalic district of the baby revealed quite peculiar lesions that led the authors to exclude that the injuries had been solely caused by violent shaking of the baby’s head, as suggested by *Shaken Baby Syndrome*. Instead, the authors hypothesised that another lesion mechanism had been added to this one, namely latero-lateral cranial compression. The comprehensive and exhaustive analysis of the case led the authors to present a new possible entity in *child abuse trauma*, namely ‘*Compressed Baby Head*’. Conclusions: To the best of our knowledge, in the current literature, no similar clinical cases have ever been described. Thus, the case’s uniqueness deserves to be brought to the attention of experts and the entire scientific community, as well as medical personnel, paediatricians, and reanimators. These professional figures are the first individuals who may encounter complex clinical cases such as the one presented in this paper; thus, they need to know how to properly manage the case and ensure protection for the abused infants and children.

## 1. Introduction

The issue of *child abuse* is of global importance, playing a crucial role in the legal, clinical, paediatric, and radiological fields [1]. The term *child abuse* identifies multiple nosological entities that have been studied and categorised over the years. Indeed, *child abuse* encompasses abusive injuries of various types and entities, including acts of violence, sexual or non-sexual, as well as cases of neglect in the care, affection, and attention of a minor [2].

In 2018, in the USA, data reported that 1770 children died due to neglect or abuse [3]. The first definition of *child abuse* came with John Caffey when the author, in 1946, studied the fractures of the long bones of a group of children and their clinical aspects in the presence of chronic subdural haematoma, thus excluding systemic pathologies capable of causing them autonomously [4].

Around 40 years ago, Ludwig and Warman first introduced the term *Shaken Baby Syndrome* (SBS), identifying a cohort of lesions and symptoms that followed the violent shaking of an infant [5]. In particular, *Shaken Baby Syndrome* strictly refers to a group of lesions and symptoms secondary to the violent shaking of the infant by parents or caregivers within the first year of life. According to the literature, SBS has a major incidence in the lower and middle social classes, and males are more affected than females in a ratio of 3:1. Several factors seem to be involved in the phenomenon of maltreatment, such as the low level of schooling of the parents or caregivers, in association with their inability to care for and provide for the child’s needs. In fact, ‘violent shaking’ mainly occurs when the child manifests difficult behaviours, such as uninterrupted crying and/or behaviour typical of childhood, that induce the parent/caregiver to make gestures of extreme violence [6].

From a pathophysiological point of view, the peculiar clinical triad is represented by subdural haematomas, cerebral oedema, and retinal haemorrhages. Despite being inherently difficult to assess, the symptoms manifested by the infant take the form of changes in sleep/wake rhythm, vomiting, convulsions, inconsolable crying, anorexia, and generalised irritability. These manifestations are usually reported to the doctor by one of the parents or the caregiver. As a result, it is very often difficult to diagnose the injuries in time and thus bring the abuse to an end.

The long-term sequelae observed are a consequence of the neurological damage caused by trauma or repeated trauma over time and are partial or total blindness, hearing loss, motor dysfunction, and intellectual disability, and in some cases, the complications are so severe that they lead to the child’s death.

Early diagnosis is based on the semiotics of the signs and symptoms shown by the infant, as well as on the indispensable assistance of imaging diagnostics; in particular, computed tomography (CT) and magnetic resonance imaging (MRI) can aid in delineating subdural haematomas and cerebral oedema, which, according to the literature, are located in the parietal and occipital areas. A further diagnostic tool is represented by the examination of the ocular fundus, which represents the gold standard for detecting retinal haemorrhage, responsible for the long-term sequelae affecting the infant’s vision. Late diagnosis can also be made by studying the neurological complications and encephalic lesions observable by MRI detecting hypoxic–ischemic encephalic areas [7,8,9,10,11,12].

However, it should be pointed out that shaking is not the only mechanism of injury described in the spectrum of *Abusive Head Trauma*; shaking is often reported in the literature to be associated with impact. Thus, a variant of SBS is defined as *Shaken Impact Syndrome*, in which the child is not simply shaken furiously but thrown violently against a surface, even if not necessarily a hard and rigid one. The impact produces a fracture or a group of focal, unilateral fractures [13].

In this case, the clinical and pathological profile presented by the infant who comes to the doctor’s attention is almost superimposable to what is observed in SBS. However, the picture is burdened by cranial, vertebral, and/or long bone fractures as a result of the infant being projected against a surface. Diagnosis by MRI and CT turns out to be the fundamental diagnostic means for detecting the lesions produced as well as their timing through the observation of reparative phenomena or relict outcomes. However, it is necessary to specify that the staging of the lesions and their correlation to a traumatic event rather than an accidental one remain one of the most critical problems in *Shaken Impact Syndrome.* As reported by Cory M. Pfeifer, vitamin D deficiency and/or genetic defects of ossification can be involved more or less significantly in the determinism of the fracture lesions, thus mystifying what has been objectified by the instrumental investigations [14,15].

However, in addition to shaking the child, violent impact against a blunt surface or the combination of the two traumatic mechanisms can generate clinical manifestations similar to SBS. For this reason, first the American Academy of Pediatrics and later the Centers for Disease Control and Prevention (CDC) defined the term abusive head trauma (AHT) to identify the set of clinical manifestations and injury mechanisms associated with abusive head injuries, judging the term to be more appropriate and precise [16,17].

The effects of AHT are catastrophic, as it is the leading cause of mortality and morbidity in children under the age of two years; AHT produces situations of marked long-term physical and intellectual disability, such as convulsions, cerebral palsy, and intellectual and learning deficits due to the severe encephalic lesions produced by the trauma, leading to death in about a third of cases [18,19,20].

The diagnosis of AHT is very complex due to the heterogeneity of the clinical and radiological signs, as well as the absence of external injuries in about 40% of cases and a generally unreliable clinical history [21]. For these reasons, a correct medico-legal, paediatric, and radiological approach is vital in order to make the correct diagnosis with the right timing, considering all the available elements, present and past, of the patient’s history [22].

Within the spectrum of clinical, radiological, and forensic manifestations related to abusive head trauma, the authors will present the case of an abused child, in which the injury modality appears to be different from those described so far in the literature and therefore cannot be categorised either as *Shaken Baby Syndrome* or *Shaken Impact Syndrome*. In particular, within the various nosological realities that fall within the spectrum of abusive head trauma, this study aims to emphasize the injury dynamics that mainly involve the cranial district.

The aim of our study is, therefore, to identify a different injury mechanism from those described in the literature, i.e., to propose a new nosological entity in the field of abusive head trauma.

## 2. Case Presentation

Nicola (fictitious name) was born in May 2013 to a healthy young woman with an excellent socio-economic background. The course of the pregnancy was regular and free of complications, and the intrauterine development of the foetus was normal and consistent with the gestational age. Delivery was eutopic and spontaneous, with no reported difficulties in the expulsive phase, facilitated by episiotomy, with the foetus in occipito-iliac-left-anterior presentation. At birth, Nicola presented an APGAR of 9 in the first minute and an ND of 10 in the fifth minute, with a negative neuromotor examination and nothing significant on objective examination.

Twenty-one days after his birth, the baby was taken by his parents to the paediatric emergency room in his hometown due to uncontrolled crying and impotence of the left upper limb. The doctors performed an X-ray of the left upper limb, which showed irregularities of the cortical profile in the proximal clavicular area, with thickening of the soft parts. The picture was compatible with a traumatic injury with active reparative phenomena.

This finding was associated with childbirth.

Ninety days after his birth, at 5 a.m., Nicola was again taken by his parents to the emergency room. During the clinical examination, the doctors observed a distressed appearance, plaintive crying, rigidity of all four limbs, pulsating bregmatic fontanelle, pale skin, and a body temperature of 38.3°.

The initial clinical suspicion was of an ongoing infection, so Nicola was transferred to the infectious diseases ward. During the transfer to the ward, the baby was seized by repeated tonic–clonic convulsions.

Diagnostic–instrumental and laboratory tests were prescribed.

Unfortunately, no information is in our possession regarding medical–legal management at the time of the event, but we acquired all the clinical records 10 years after the event.

### 2.1. Clinical Presentation

Nicola underwent a thorough, objective examination, which revealed the presence of multiple small ecchymotic lesions in different body regions: frontal, nasal, right and left cheek, chin region, right nipple, left gluteal region, and fifth finger of the right hand. The lesions mentioned above were described and documented with different colour stages, probably occurring at different times.

The infant underwent a specialist ophthalmic examination, which objectified the presence of diffuse oedema of the retina on the right, extending to the posterior pole, with congestion of the venous vessels. On the left, there was modest oedema of the upper eyelid and bulbar conjunctiva, as well as the presence of intraretinal and preretinal haemorrhages, some loculated, with no visualisation of the optic papilla, macula, and ipsilateral fovea.

### 2.2. Radiological Features

The cranial CT scan performed immediately after hospital admission showed the presence of diffuse cerebral oedema, as well as two lesion complexes:

First lesion complex (Figure 1) at the level of the left parietal bone, presence of a cranial fracture with a linear course, starting from the sagittal suture, with craniocaudal direction. Extensive subgaleal haematoma in the parieto-occipital area.

Second lesion complex (Figure 2) at the level of the right parietal bone, presence of a fracture complex consisting of three radiating fractures branching off from the ‘point of impact’ located at the level of the right parietal draft:-Inferior “branch” (A), with a linear course, extending cranio-caudally and involving the internal cranial plateau and the full-thickness diploe;-Upper “branch” (B), linear course, extending caudo-cranially, to the left, terminating at the posterior third of the sagittal suture;-Posterior “branch” (C), linear course, extending antero-posteriorly, ending at the lambdoid suture.

Taken together, the three fracture “branches” described above, together with the sagittal suture and the right branch of the lambdoid suture, identified the structure of a quadrangular ‘baseball diamond’-like dowel.

A right parietal subgaleal haematoma and a subdural and subarachnoid haemorrhage in the homolateral fronto-parietal-temporal area were also revealed (Figure 3).

The appearance and extent of the haematomas were highlighted bilaterally, and the characteristics of the radiological images suggested that they may have appeared at the same time.

The newborn underwent a control brainstem and encephalic MRI on the same day. This investigation, in addition to confirming the presence of the right and left subgaleal haematomas already shown by the cranial CT scan, highlighted the presence of diffuse cerebral oedema, a right parietal subdural haematoma, subdural haematic effusion localised between the sickle and tentorium of the cerebellum, epidural haemorrhage at the posterior cerebellar site, intraparenchymal cortical haemorrhage at the left temporal site, frontobasal subarachnoid haemorrhage bilaterally, and hygromatous stratum at the left frontotemporal site.

The infant also underwent a chest X-ray, which showed the presence of multiple rib fractures and a left clavicular fracture. The lower limb X-ray documented the presence of a tibial diaphyseal spiral fracture bilaterally undergoing ossification, with the presence of bony callus more significant on the left tibia (Figure 4).

Subsequent radiological investigations carried out during the infant’s stay in the ICU did not reveal any presence of vertebral and/or spinal cord lesions. On the contrary, they confirmed the evolution of the cranio-encephalic lesion complex, attesting to the appearance of diffuse ischaemic phenomena at the encephalic level, as discussed below.

### 2.3. Neurological Outcome

Subsequent instrumental investigations were performed on the baby to assess cranio-encephalic outcomes. In particular, the MRI (Figure 5) performed about a month after the presumed trauma revealed a picture characterised by cortical atrophy and cystic evolution of the brain parenchyma, mainly in the occipital region bilaterally.

### 2.4. Genetic Investigations

Following the preliminary clinical investigation, the medical personnel on duty suspected infant maltreatment and therefore informed the judicial authority. The court ordered the immediate removal of the newborn from the family and relocated him into foster care.

In the months and years that followed, the newborn was subjected to numerous wide-ranging genetic investigations to exclude any possible malformations and/or bone changes related to systemic diseases or genetic syndromes, which could have been the cause of the numerous bone fractures found.

In particular, a molecular analysis of the COL1A1 and COL1A2 genes, responsible for the pathogenesis of osteogenesis imperfecta, was immediately carried out, which yielded negative results.

Nicola was then subjected to CGH array analysis [23,24], which revealed the presence of a 5 Mb microduplication of the long arm of chromosome 14 of uncertain significance. A similar chromosomal aberration was also found in the genetic analysis performed on the newborn’s father in the absence of a noteworthy clinical finding.

The study of the genetic data acquired excluded a correlation between the microduplication found in Nicola’s DNA and his clinical picture, characterised by multiple-region fractures.

The baby was also subjected to exome analysis by next-generation sequencing (NGS) [25,26]. This investigation was performed on the assumption that most of the modifications responsible for hereditary and rare diseases are located in the exome. Thus, the results excluded the presence of DNA variants of possible pathogenic significance in the 4800 genes analysed.

In conclusion, the genetic investigations on baby Nicola allowed the authors to exclude the possibility that the immediate and subsequent severe cranial and neuropsychic injury and multidistrict fracture could be related to a genetic origin.

Overall, the injury pattern observed in the newborn involved different body districts, with a more significant predominance of trauma in the cranium (Table 1).

## 3. Discussion

The analysis of this case study allowed us to better investigate the topic of *child abuse*. Referring to the definition given by the World Health Organization in 2002, ‘Child abuse or maltreatment constitutes all forms of physical and/or emotional ill-treatment, sexual abuse, neglect or negligent treatment or commercial or other exploitation, resulting in actual or potential harm to the child’s health, survival, development or dignity in the context of a relationship of responsibility, trust or power’ [27].

Within this context, multiple terms have been used over the years to define child maltreatment, such as ‘battered child syndrome’, ‘parent infant traumatic stress syndrome’, and ‘shaken baby syndrome’, but the most preferred term since 2009 is ‘abusive head trauma’. To diagnose one of these syndromes, it is necessary to subject the child to a thorough, objective examination and a targeted instrumental investigation. Of great importance is the role played by instrumental investigations such as CT and MRI as well as fundus oculi and retinal examination [28,29].

Although diagnostic criteria and helpful advice for the diagnosis and the treatment of such syndromes are widely described in the literature, in clinical and forensic practice, there may be more complex cases of difficult approaches. This is determined by the presence of heterogeneous and non-specific clinical and radiological data and, above all, the absence in 40% of cases of external injuries on the child’s body, which could better guide the diagnosis of *child abuse* in general or the diagnosis of *Shaken Baby Syndrome* or *Shaken Impact Syndrome* [21,22,30].

Furthermore, excluding alternative causes that may have led to a compromised systemic picture in the child, with the presence of injuries that may mimic ecchymoses and/or traumatic fractures, is of considerable importance. It is worth remembering that there are specific systemic pathologies, including metabolic ones, that can mimic the lesions typical of *Abusive Head Trauma*: vitamin D deficiency, metabolic disorders, skeletal pathologies (osteogenesis imperfecta, Menkes disease, and osteopetrosis), infections, glutaric aciduria type I, bleeding diathesis, connective tissue diseases and vasculopathies, Wormian bones, and congenital skull depression [22].

The American Academy of Pediatrics (AAP) has established guidelines in cases of suspected *Abusive Head Trauma*, including routine laboratory tests useful for studying coagulation as well as neonatal screening tests. The same guidelines also recommend cranial CT and MRI imaging of the brain and of the vertebral column [16]. Another eventuality to be considered is the presence of signs and symptoms of postcranial physical abuse, with associated occult cranial and spinal injuries [31].

For all these reasons, and because of the heterogeneity of clinical presentations, the Royal College of Radiologists (RCR) of the United Kingdom recommends the performance of instrumental cranial CT examination in every case of suspected child abuse under one year of age. In particular, if there is any evidence of intracranial haemorrhage, parenchymal injury, or skull fracture on the CT scan, the recommendation is to perform an MRI scan within 2–5 days. In case of a negative CT scan but in the presence of ongoing neurological symptoms, an MRI scan is still recommended [32].

Moreover, it is essential to combine clinical and instrumental studies with genetic investigations in order to search for or exclude any genetic pathologies that may induce spontaneous fractures or lead to lesions following minor trauma, such as the genetic study involved in osteogenesis imperfecta [33].

In the case study presented here, the child victim presented the characteristic symptoms and clinical signs of *Shaken Baby Syndrome*, i.e., the classic triad of subdural haematoma, cerebral oedema, and retinal haemorrhage, associated with more non-specific clinical manifestations such as subgaleal haematomas, multiple rib fractures, and long bone fractures [34].

*Shaken Baby Syndrome* (SBS), also known as *Abusive Head Trauma* (AHT) [35], represents a form of child maltreatment and consists of the violent shaking of a child. This mechanism results in the production of encephalic trauma, as well as major neurological sequelae, disability, or death [36,37].

However, in the case presented by the authors, the presence of a typical cranial lesion characterised by two symmetrical fracture complexes on the parietal bones was highlighted. This picture neither fulfils the diagnostic criteria for *Shaken Baby Syndrome*, given the presence of fracture lesions, nor can it be classified as *Shaken Impact Syndrome* due to the two symmetrical fracture complexes observed.

The instrumental investigations performed on the child (a CT scan and an MRI scan) showed the presence of three radiating fractures on the right parietal bone, one inferior, one superior, and one posterior, which depart from the same point, defined as the ‘point of impact’. Finally, the shape and direction of the three branches, together with the sagittal suture and the lambdoid suture, made it possible to identify a quadrangular dowel resembling the shape of a ‘baseball diamond’.

At the left parietal bone level, however, instrumental investigations revealed the presence of a single linear fracture rhyme, independent from the previous fracture pattern.

Firstly, the timing of the aforementioned lesions was studied, which was possible thanks to the analysis and comparison of the CT and MRI images taken at the time of the newborn’s admission to the hospital when he was under the care of the first doctors who visited him. Based on these data, it was possible to assume that the cranio-encephalic traumatism detected on the newborn child dated back to the hours immediately prior to admission to the hospital.

In this sense, according to the objective characteristics of the ophthalmological instrumental investigations, it was possible to confirm that the retinal pathological complex, characterised by the presence of retinal oedema and haemorrhages, sometimes loculated, was also to be understood as occurring at the same time as the cranio-brain lesions, which means in the hours immediately preceding the baby’s admission to the hospital.

These findings were considered as ‘abnormal’ for the diagnosis of *Shaken Baby Syndrome*, leading the authors to hypothesise that the injury mechanism responsible for the trauma observed was not just ‘simple’ shaking of the child.

The presence, in fact, of cranial fracture complexes led the authors to hypothesise that the injury mechanism comprehended a violent cranial impact on a hard surface, as typically observed in *Shaken Impact Syndrome*. In the *Abusive Head Trauma* context, it is possible to speak of *Shaken Baby Syndrome* by referring to a shaking mechanism of the child or an impact of the latter or a combination of the two traumatic actions, which will result in neurological injuries. (Figure 6).

In the case presented, it was first hypothesised that a violent succussion had occurred, particularly at the latero-lateral cranial side of the infant, from left to right, otherwise induced, with cranial impact against a flat, smooth, and resistant surface localised in the right parietal area.

However, the in-depth study of the characteristics of the lesion complexes found in the child led the authors to hypothesise a new, different lesion mechanism, other than *Shaken Baby Syndrome* and *Shaken Impact Syndrome*.

The presence, in fact, of two symmetrical fracture complexes on the parietal bones, one of which was larger (on the right) and one of which was smaller (on the left), led the authors to assume that this injury was produced by a double coeval impact, from both cranial surfaces, right and left, caused by a compressive action at the cranial level applied laterally.

In particular, it was hypothesised, considering the site and the different entities of the fractures present, the associated subgaleal haematomas, more present on the right, as well as the absence of ecchymotic skin lesions at the cranial level, patterned or not, that the compression was determined by the action of a hand positioned on the left parietal bone, thus coherent with the linear fracture highlighted by the instrumental investigations. The compression, according to the authors, would therefore have been exerted from the left parietal bone towards the right parietal bone, which was resting on a rigid surface, hence where the lesion complex was most evident and where a point of departure of the three radiating fractures, called the ‘point of impact’, could be clearly observed. This is also confirmed by the radiological appearance and extent of the subgaleal haematomas found bilaterally at the cranial level, which allows us to affirm the contemporaneity of the lesions, which were therefore determined by a single traumatic mechanism.

Furthermore, the predominance of fracture injuries over the shaking-related ones observed on the infant justifies the absence of vertebral injuries due to the discharge of kinetic energy acquired by the infant’s head impact on the right parietal bone.

On the other hand, with regard to the spiral fractures of the tibiae and the multiple costal fractures, the presence of bone callus on radiographic images made it possible to define the lesions as non-coincident with cranio-encephalic trauma. In particular, the tibial fractures, due to the particular spiral conformation, were not correlated with direct traumatism but due to a mechanism of extreme rotation of each lower limb. The detected rib fractures, located at the level of the posterior arch of two adjacent ribs, are also typically frequent in cases of *child abuse* [38]. For all these reasons, the authors identify the pathogenetic mechanism on the basis of the head injury observed in the child as a case of *‘Compressed Baby Head’* (Figure 7).

The lack of similar clinical cases in the literature, together with the description of a complex case study that can be well explained by the injurious mechanism described in this paper, albeit with the limitations due to the singularity of the case, is intended to be a cause for reflection for the entire scientific community.

*Child abuse* cases are always worthy of research and investigation. In the current case, the authors wanted to present a new possible injury entity to be considered in the *Abusive Head Trauma* field, namely ‘*Compressed Baby Head*’.

The diagnostic criteria proposed for the definition of ‘*Compressed Baby Head*’ are:-Bilateral skull fractures or fracture complexes;-The presence of bilateral perifractural cranial haematomas and/or haemorrhages, with similar characteristics in terms of extent and timing;-Cerebral oedema;-Retinal haemorrhage.

We also reiterate the fundamental importance of performing targeted radiological investigations: cranial CT is to be performed immediately, especially if child maltreatment is suspected. Brain magnetic resonance imaging is also recommended to be performed in the event of a positive cranial CT scan and in the following days in order to adequately monitor the evolution of fracture lesions and haemorrhagic lesions.

## 4. Conclusions

*Child abuse* is, unfortunately, a topical subject. Diagnosing child maltreatment is oftentimes incredibly challenging and requires the attention and experience of the medical professionals examining the child. Hypothesising the dynamics that may have determined the injuries observed on a child is of considerable importance to distinguish intentional from accidental trauma (*non-Abusive Head Trauma*). The medical and forensic literature is of great help in identifying and recognizing even the most complex cases of child abuse, and the guidelines offer important help and recommendations to properly handle the case and the abused child.

However, some clinical cases can be complex and atypical. In this paper, the authors presented a case of *Abusive Head Trauma* characterised by a double lesion complex at the cranial level, with symmetrical and bilateral localisation at the parietal bones. Through the study and analysis of the injuries found, the authors explained the cranial injury mechanism that may have determined the traumatic lesions observed in the child. The proposed mechanism is of compressive nature, thus identifying a new possible clinical entity included in *Abusive Head Trauma*: the ‘*Compressed Baby Head*’.

By presenting this case, the authors wish to propose to the scientific community that this new type of traumatic event should be included in the context of *Abusive Head Trauma*.

## Figures and Tables

**Figure 1 children-10-01003-f001:**
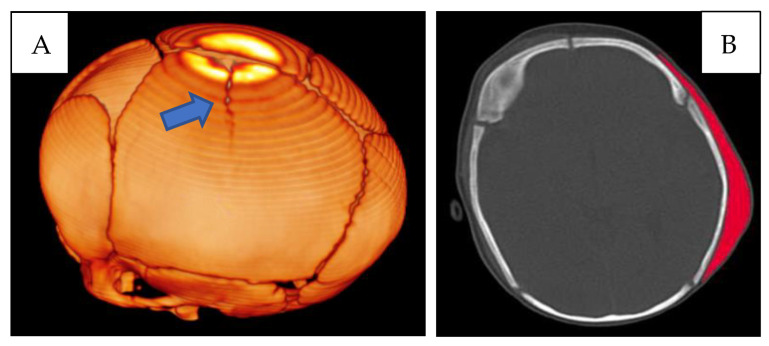
(**A**) CT skull—3D reconstruction: fracture with a linear course in the left parietal location (blue arrow). (**B**) CT skull, subgaleal haematoma in the parieto-occipital site (red).

**Figure 2 children-10-01003-f002:**
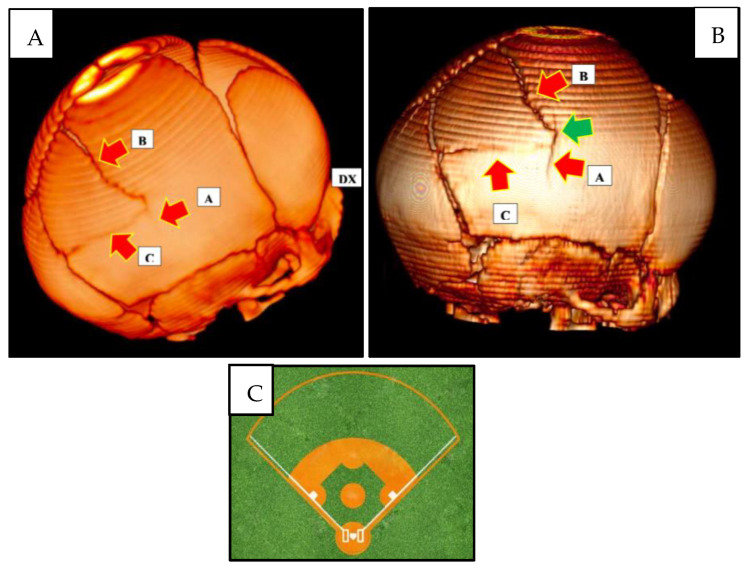
Three-dimensional skull CT reconstruction. (**A**) The red arrows indicate the three fracture branches: ‘A’ lower branch; ‘B’ upper branch; and ‘C’ posterior branch. ‘DX’ indicates the right side. (**B**) The red arrows indicate the three fracture branches: ‘A’ lower branch; ‘B’ upper branch, and ‘C’ posterior branch. The green arrow indicates the point of cranial impact. (**C**) ‘Baseball diamond’ image.

**Figure 3 children-10-01003-f003:**
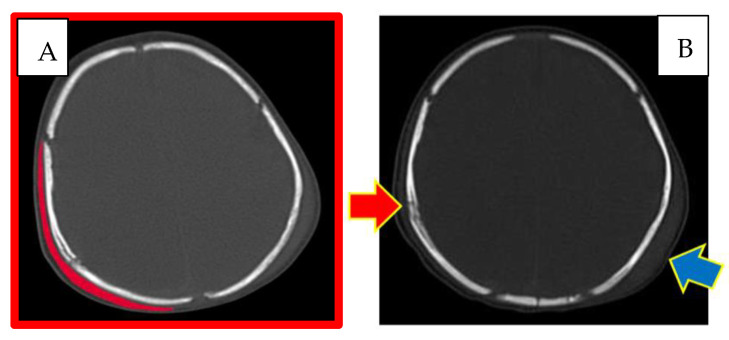
CT skull. (**A**) Right subgaleal haematoma. (**B**) Red arrow: right parietal bone diploe involvement and consensual subgaleal haematoma; blue arrow: left subgaleal haematoma.

**Figure 4 children-10-01003-f004:**
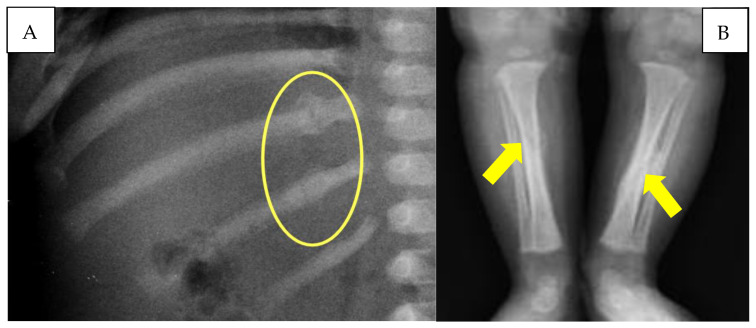
(**A**) Rib fractures with reparative phenomena (yellow circle). (**B**) Diaphyseal tibial spiral fractures (yellow arrows).

**Figure 5 children-10-01003-f005:**
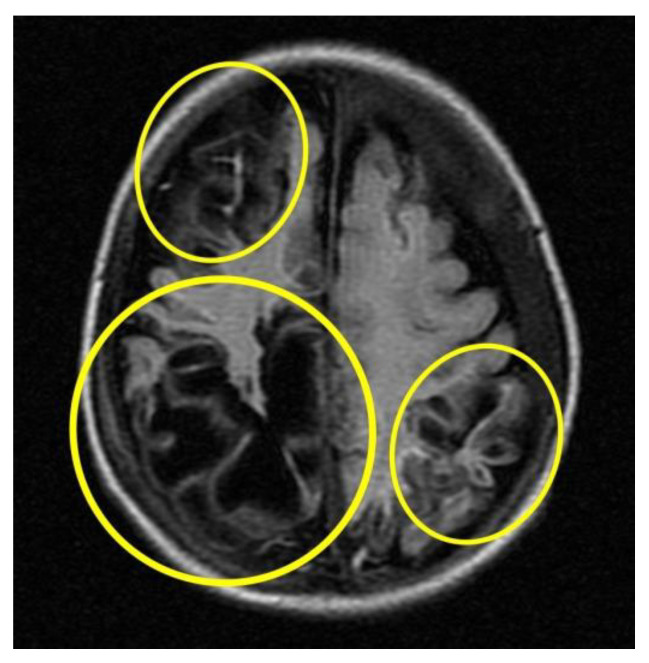
MRI cortical atrophy and cystic evolution of the brain parenchyma (circles in yellow).

**Figure 6 children-10-01003-f006:**
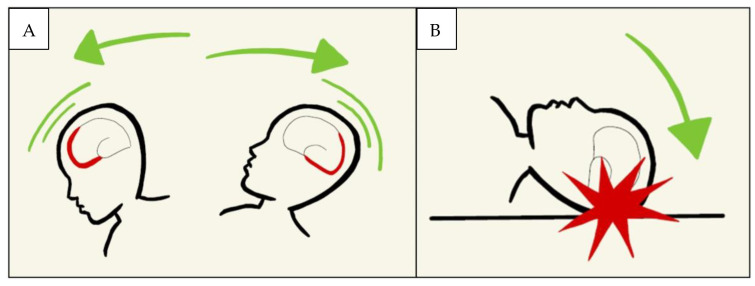
(**A**) Shaken baby syndrome. (**B**) Shaken impact syndrome.

**Figure 7 children-10-01003-f007:**
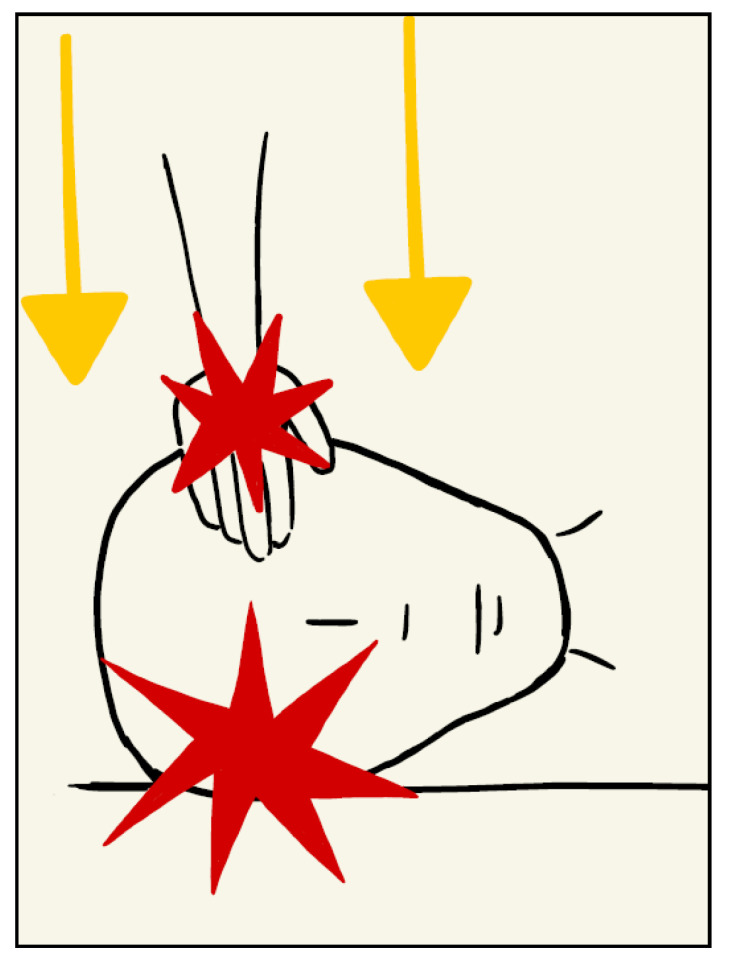
*Compressed Baby Head*.

**Table 1 children-10-01003-t001:** The injuries found on the child broken down by districts: head injuries, eye injuries, and other injuries.

Cranial Injuries	Eye Injuries	Other Injury
- no. 3 right parietal bone fractures - Right subgaleal haematoma - Subdural haemorrhage and subarachnoid in the right fronto-parieto-temporal area - Left parietal fracture - Left subgaleal haematoma - Diffuse cerebral oedema	- Right eye diffuse retinal oedema - Left eye: oedema of the upper eyelid and bulbar conjunctiva - Left eye: intraretinal and preretinal haemorrhages, some loculated	- Left clavicle fracture - Multiple rib fractures - Spiral fractures of the bilateral tibial diaphysis

## Data Availability

Data sharing not applicable.

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
