# Peer review of "‘Compressed Baby Head’: A New ‘Abusive Head Trauma’ Entity?"

_children, 2023, doi:10.3390/children10061003_

Round 1

Reviewer 1 Report

Dear authors,

The manuscript, which is titled" Compressed Baby Head': a new 'Abusive Head Trauma' entity?" is related to the case of abuse by their parents and was well written. It is interesting to read, and also, there is a holistic approach in the paper, both medical and legal. I appreciate your effort to write and collect the data and history of this case. I have some recommendations related to this paper;

-Nearly all paragraphs needed to be longer. Remember the paragraph should consist of at least three sentences. 

- MRI and CT should be written without abbreviation in the first place in the text. 

I wish you success in your work. 

Author Response

Dear Reviewer 1,

thank you for your review and recommendations.

  1. I will adjust the length of paragraphs
  2. I will enter the meaning of the abbreviations MRI and CT.

I will make the changes you recommended. Thank you!

Reviewer 2 Report

1.For Figure 1, the letters A and B in the pictures were not clear.

2.Line 293  the end of the word “trauma” has a symbol “1” ?

3. The authors presented a worthy case report, and the authors maybe follow the “CARE guidelines” to make the report more significant.

Author Response

Dear Reviewer 2,

thank you for your review and recommendations.

  1. I will make the letters A and B in Figure 1 clearer.
  2. Thank you for pointing this out, is this a typo.
  3. Thank you for the recommendation, we followed the CARE checklist and included also the informed consent from the legal protector.

Thank you!

Reviewer 3 Report

In this study, the concept of "Compressed Baby Head" was discussed in detail with a case study and guidance was given to pediatricians, emergency physicians and health professionals. The article is written in an appropriate language and is a successful study. It is capable of contributing to the literature on child abuse.  There are some minor suggestions for correction.

1. Information about the characteristics, precursors and personalities of the perpetrators who cause this event in children should be added and psychosocial aspects should be discussed.

2. The medical-legal aspects of these processes should be mentioned and brief information about what has been done in the context of this case should be added.   

Author Response

Dear Reviewer,

thank you for your review and recommendations.

  1. In our paper, we have chosen not to include information regarding the perpetrators who caused the mistreatment because the criminal trial is still ongoing. For this reason, we wanted to avoid addressing this topic and focused only on the mechanism that generated the injuries to the child.
  2. We also consider this issue to be of great importance. Unfortunately, no information is in our possession regarding medical-legal management at the time of the event, but we acquired all the clinical records 10 years after the event. However, we added this specification in the text.

Thank you!